# Kramers–Kronig Relation for Attenuated Total Reflection from a Metal–Dielectric Interface Where Surface Plasmon Polaritons Are Excited

**DOI:** 10.3390/nano11113063

**Published:** 2021-11-14

**Authors:** Heongkyu Ju

**Affiliations:** Department of Physics, Gachon University, Seongnam-si 13120, Korea; batu@gachon.ac.kr

**Keywords:** surface plasmon resonance, optical dispersion, reflection spectrum, Kramers–Kronig relation, causality, phase dispersion, optical density

## Abstract

The applicability of the Kramers–Kronig relation for attenuated total reflection (ATR) from a metal–dielectric interface that can excite surface plasmon polaritons (SPP) is theoretically investigated. The plasmon-induced attenuation of reflected light can be taken as the resonant absorption of light through a virtual absorptive medium. The optical phase shift of light reflected from the SPP-generating interface is calculated using the KK relation, for which the spectral dependence of ATR is used at around the plasmonic resonance. The KK relation-calculated phase shift shows good agreement with that directly obtained from the reflection coefficient, calculated by a field transfer matrix formula at around the resonance. This indicates that physical causality also produces the spectral dependence of the phase of the leakage field radiated by surface plasmons that would interfere with the reflected part of light incident to the interface. This is analogous with optical dispersion in an absorptive medium where the phase of the secondary field induced by a medium polarization, which interferes with a polarization-stimulating incident field, has a spectral dependence that stems from physical causality.

## 1. Introduction

The refraction of light that propagates in a medium depends on its wavelength or frequency: a phenomenon known as the refractive index dispersion. It becomes more conspicuous when the light wavelength approaches a spectral band of the resonance of optical absorption, the so-called anomalous dispersion region, where the second derivative of the wavenumber with respect to frequency turns negative [1,2].

The index dispersion that occurs in a linear passive medium can basically be understood by the Kramers–Kronig relation (KK relation) [1,2,3]. Mathematically, it consists of the cross-coupled integral equations in a frequency domain, which connect between the real and imaginary parts of an electric susceptibility of a medium through which light propagates, i.e., $\chi^{\prime }\left(\omega \right)$ and $\chi^{\prime \prime }\left(\omega \right)$, respectively. This implies that the frequency dependence of light absorption dictates the frequency dependence of light refraction or vice versa. The KK relation can be derived using physical causality—the medium cannot react prior to disturbance—that leads to the convolution theorem in a linear passive medium. Thus, this produces the linear relation between a response function, $\chi \left(\omega \right)\left[=\chi^{\prime }\left(\omega \right)+i\chi^{\prime \prime }\left(\omega \right)\right]$ and an applied electric field E(ω) in a frequency domain. The causality-supported analyticity of χ(ω) as a function of a complex frequency eventually provides the mathematical form of the KK relation [1,2,3].

From the field interference point of view, light absorption during propagation through a medium occurs as a result of a phase mismatch between an applied electric field and its induced medium polarization (electric dipole moments per unit volume) with its dephasing degree determined by the relative ratio of $\chi^{\prime \prime }\left(\omega \right)$ with respect to $\chi^{\prime }\left(\omega \right)$ [1,3]. Since such a dephasing degree depends on the light frequency under physical causality, optical absorption becomes frequency-dependent. Consequently, this makes the medium dispersive where a refractive index depends on a frequency of light via the KK relation.

Surface plasmon polariton (SPP) is the resonance mode resulting from coherent coupling between the electromagnetic fields of transverse magnetic (TM) polarization and collective oscillation of surface electrons at a metal–dielectric interface. SPP can be excited by injecting right photon momenta in parallel to the surface via the evanescent coupling of an incident light into the interface, typically in a Kretschmann–Raether configuration [4,5,6,7], in optical fibers [8,9,10,11] or via grating couplers that modulate the surface parallel photon momenta [12,13,14]. In a wavelength interrogation method, the optical reflectance from the metal–dielectric interface reduces significantly at a narrow spectral band due to the SPP excitation at the surface, the so-called attenuated total reflection (ATR). The use of plasmons can allow the sub-diffraction-limited confinement of optical fields with enhanced field strength [15], and gradient optical fields on a subwavelength scale [16] and find its extension to magnetic plasmons for a refractive index sensing [17].

In a point of optical energy carried by light, one may regard the frequency dependence of the reflected light intensity at around the plasmon resonance as the frequency dependence of the intensity of transmitted light through a virtual absorptive medium at around the absorption resonance. Based on this assumption, this work theoretically investigates the applicability of the KK relation for ATR from a metal–dielectric interface in a Kretschmann–Raether configuration. The KK relation is used to theoretically derive the frequency-dependent phase shifts of reflected light using a frequency-dependent reduction in reflectance that is calculated with a field transfer matrix formula (Abelle matrix) [18,19,20]. The KK relation-derived phase shifts are then compared with those directly calculated with the transfer matrix method, with discussions in the context of the phase shift dispersion associated with the spectral dependence of light attenuation.

## 2. ATR in a Kretschmann–Raether Configuration

Figure 1 shows the schematic of a Kretschmann–Raether configuration assumed for calculating the reflection coefficient r(ω) as a function of an optical frequency ω. A polychromatic light source of TM polarization is assumed to be incident to a metal–dielectric interface at a given incident angle ($\theta_{in}$) to interrogate the wavelengths for surface plasmon resonance. The interface between air and the 50 nm-thick gold (Au) film is used to generate the surface plasmon polaritons at a certain spectral band. This plasmon excitation produces a reduction in reflectance $R\left(\lambda \right)\equiv {\left|r\left(\lambda \right)\right|}^{2}$, which depends strongly on the wavelength (λ) of light at around its resonance.

The transfer matrix derived expression for r(λ) is given for TM polarization as follows [18,19,20]:(1)r(λ)=[m11Auγp+m12AuγpγA−m21Au−m22AuγA][m11Auγp+m12AuγpγA+m21Au+m22AuγA]
(2)t(λ)=[2γAϵrp/ϵrA][m11Auγp+m12AuγpγA+m21Au+m22AuγA]
where $m_{ij}^{\mathrm{Au}}$ is the element of the 2×2 field transfer matrix as given by:(3)$$m^{\mathrm{Au}}=\left(\begin{array}{cc} \cos\delta & \mathrm{isin}\delta /\gamma_{\mathrm{Au}}\\ i\gamma_{\mathrm{Au}}\sin\delta & \cos\delta \end{array}\right)$$

The layer parameters for a prism (N-BK7 prism), the Au film, and air, are given by $\gamma_{p}=\left(\omega \epsilon_{r}^{p}/k_{z}^{p}c^{2}\right),$ $\gamma_{\mathrm{Au}}=\left(\omega \epsilon_{r}^{\mathrm{Au}}/k_{z}^{\mathrm{Au}}c^{2}\right),$ and $\gamma_{A}=\left(\omega \epsilon_{r}^{A}/k_{z}^{A}c^{2}\right).$ $k_{z}^{p}$, $k_{z}^{\mathrm{Au}}$, and $k_{z}^{A}$ are the z-components of the propagation vector of light in the layers of the prism, the Au film and air, respectively. Here $\delta =k_{z}^{\mathrm{Au}}d_{\mathrm{Au}}$ is the complex optical phase shift induced by the Au film of the thickness $d_{\mathrm{Au}}$. $\epsilon_{r}^{p}$, $\epsilon_{r}^{\mathrm{Au}}$, and $\epsilon_{r}^{A}$ are the relative permittivities of the layers of the prism, the Au film and air, respectively. c is the speed of light in vacuum.

It is noted that $k_{z}^{\mathrm{Au}}$ is wavelength-dependent, as is $\epsilon_{r}^{\mathrm{Au}}$, making r strongly wavelength-dependent, particularly at around the plasmonic resonance. The wavelength-dependent $\epsilon_{r}^{\mathrm{Au}}\left(\lambda \right)={\left[n\left(\lambda \right)+i\kappa \left(\lambda \right)\right]}^{2}$ is given by n(λ) and κ(λ) over the wavelength range of 547.8–676.0 nm as follows:(4)$$n\left(\lambda \right)=A_{0}+A_{1}\left(10^{9}\lambda \right)+A_{2}{\left(10^{9}\lambda \right)}^{2}+A_{3}{\left(10^{9}\lambda \right)}^{3},$$
(5)$$\kappa \left(\lambda \right)=B_{0}+B_{1}\left(10^{9}\lambda \right)+B_{2}{\left(10^{9}\lambda \right)}^{2},$$
where $A_{0}\;=\;24.0199$, $A_{1}$ = −0.10528, $A_{2}\;=\;1.55802\times 10^{-4}$, $A_{3}\;=\;-7.73282\times 10^{-8}$, and $B_{0}\;=\;-8.43082$, $B_{1}$ = 0.0272, $B_{2}\;=\;-1.29539\times 10^{-5}$. These parameters are obtained by a polynomial fit to data of refractive indices n(λ) and extinction coefficients κ(λ) for Au films which are available from [21,22].

Figure 2A shows the reflectance spectrum, i.e., $R\left(\lambda \right)\equiv {\left|r\left(\lambda \right)\right|}^{2}$ at around the plasmonic resonance for a given incident angle of $43.85^{o}$, while Figure 2B is the natural logarithm of 1/R(λ). This spectrum shows the strong spectral dependence of attenuation of light reflected from the Au film–air interface at around the plasmonic resonance that occurs at the wavelength of about 615 nm.

The phases of the complex reflection and transmission coefficients, i.e., r(λ) and t(λ), are calculated using Equations (1) and (2), as shown in Figure 3. These phases represent the relative phase shift of the reflected and the transmitted electric fields with respect to that of an incident field.

## 3. Phase Dispersion Calculation with Kramers–Kronig Relation

Let us consider the attenuated intensity of reflected light as the attenuated intensity of light transmitted through a virtual absorptive medium. The reflectance can thus be taken as the transmittance T, enabling $\log_{e}\left[1/R\right]$ to be treated as absorbance or optical density, $\mathrm{OD}=\log_{e}\left[1/T\right]$ for the virtual medium.

In general, physical causality dictates that the complex function $\chi \left(\omega \right)=\chi^{\prime }\left(\omega \right)+i\chi^{\prime \prime }\left(\omega \right)$ (as a function of a real frequency) requires a pair of the cross-coupled integral equations, i.e., the KK relation as follows [1,2,3]:(6)$$\chi^{\prime }\left(\omega \right)=\frac{2}{\pi }P\int_{0}^{\infty }\frac{\chi^{\prime \prime }\left(\omega \right)\omega^{\prime }}{{\omega^{\prime }}^{2}-\omega^{2}}d\omega^{\prime }$$
(7)$$\chi^{\prime \prime }\left(\omega \right)=-\frac{2\omega }{\pi }P\int_{0}^{\infty }\frac{\chi^{\prime }\left(\omega^{\prime }\right)\omega^{\prime }}{{\omega^{\prime }}^{2}-\omega^{2}}d\omega^{\prime }$$
where P denotes the Cauchy principal value.

One can consider the relative permittivity of the virtual medium to be divided into two parts, those from the non-resonant and resonant contributions, i.e., $\epsilon_{r}^{v}=\left(1+\chi_{NR}\right)+\Delta \chi_{R}$. Here, $1+\chi_{NR}$ denotes the non-resonant part with the non-resonant real susceptibility $\chi_{NR}$, while $\Delta \chi_{R}\left(=\Delta \chi_{R}^{\prime }+i\Delta \chi_{R}^{\prime \prime }\right)$ denotes the complex susceptibility due to a resonant absorption. The condition $\left|\Delta \chi_{R}/\left(1+\chi_{NR}\right)\right|\ll 1$ enables the resonant contribution to the real and imaginary parts of the complex refractive index to be expressed as:(8)$$\Delta n_{R}=\frac{\Delta \chi_{R}^{\prime }}{2n_{NR}}\;,$$
(9)$$\Delta \kappa_{R}=\frac{\Delta \chi_{R}^{\prime \prime }}{2n_{NR}},$$
where $n_{NR}={\left(1+\chi_{NR}\right)}^{1/2}$ denotes the refractive index due to the non-resonant response of the virtual medium. Equations (6) and (8) then give one of the equations for the Kramers–Kronig relation under a resonant absorption, which relates the resonant absorption coefficient, i.e., $\alpha_{R}\left(\omega \right)$ and its corresponding contribution to the real part of the refractive index [23,24], as given by:(10)$$\Delta n_{R}\left(\omega \right)=\frac{c}{\pi }P\int_{0}^{\infty }\frac{\alpha_{R}\left(\omega \right)}{{\omega^{\prime }}^{2}-\omega^{2}}d\omega^{\prime },$$

Here, $\alpha_{R}\left(\omega \right)$ represents the resonant absorption due to the virtual medium. Assume that the virtual medium has a certain thickness through which light propagates. Multiplying both sides of Equation (10) by the propagation distance (thickness) and a vacuum wave number, one can obtain, on the left-hand side of Equation (10), the corresponding phase shift through the thickness. On the right-hand side, the product of $\alpha_{R}\left(\omega \right)$ and the propagation distance can then be denoted by the optical density OD(ω) (or absorbance), as given by:(11)$$\Delta \phi_{\mathrm{KK}}\left(\omega \right)=\frac{\omega }{\pi }P\int_{0}^{\infty }\frac{\mathrm{OD}\left(\omega^{\prime }\right)}{{\omega^{\prime }}^{2}-\omega^{2}}d\omega^{\prime }$$

An alternative expression for the phase shift as a function of λ is also given by:(12)$$\Delta \phi_{\mathrm{KK}}\left(\lambda \right)=\frac{\lambda }{\pi }P\int_{0}^{\infty }\frac{\mathrm{OD}\left(\lambda^{\prime }\right)d\lambda^{\prime }}{\lambda^{2}-{\lambda^{\prime }}^{2}}$$

We reiterate that that the optical density (or absorbance) as a function of λ can be obtained from the reflectance spectrum calculated using the field transfer formula, as shown in Figure 2B. Substituting the calculated OD(λ) into Equation (12) eventually provides the phase shift computed by the KK relation. This phase shift spectrum, which has an asymmetric nature across the resonance, spans from negative to positive values at around the central wavelength $\lambda_{c}≅$ 615 nm where the reflectance dip also occurs (see Figure 2A). The phase offset ($\phi_{0}=-2.14$ radian) was then chosen to be added to $\Delta \phi_{\mathrm{KK}}\left(\lambda_{c}\right)$ in order to ensure the following equality:(13)$$\Delta \phi_{\mathrm{KK}}\left(\lambda \right)\;+\phi_{0}=\;\mathrm{phase}\;\mathrm{of}\;r\left(\lambda \right)\;\mathrm{at}\;\lambda =\lambda_{c},$$
where the phase of r(λ) is calculated above (see solid curves in Figure 3).

Figure 4 shows the comparison between the offset-added $\Delta \phi_{\mathrm{KK}}$ and the phase of r(λ). Good agreement between the two phases is obtained particularly at around the resonance wavelengths. For other incident angles ($\theta_{in}$) of light, such good agreements are also found at around the resonance wavelengths that shift with $\theta_{in},$ as shown in Figure 5A–D.

It is known that, in general, a wavelength-independent phase shift that can model the phase offset can occur during light propagation through a transparent medium (off an absorption resonance). This fact leads one to model the SPP excitation system by two adjacent virtual layers comprised of a resonantly absorptive layer and a transparent layer. The former layer derives the wavelength dependence of a phase shift (refractive index) from the resonant absorption, while the latter derives the wavelength-independent $\phi_{0}$ from its transparence.

At around resonant wavelengths, the analogy of optical dispersion between transmitted light through an absorptive medium and reflected light from an SPP-generating interface can also be understood by the wavelength dependence of a certain degree of dephasing between an incident wave and a secondary wave generated by a responding medium (i.e., an absorptive medium or an SPP-generating interface). The secondary waves produced by a responding medium are collinear with the incident waves, producing interference between the two. The dephasing degree that governs the intensity attenuation of the resultant waves of interference then depends on a wavelength of an incident light in both cases, i.e., the cases of an absorptive medium and an SPP-generating interface. Eventually, the phase shift of the attenuated light is wavelength-dependent, as is its intensity.

In the former case, the incident field-induced medium polarization can be decomposed into real and imaginary parts with each part being wavelength-dependent. Consequently, the relative ratio of the imaginary part with respect to the real part can account for the wavelength-dependent phase mismatch between a secondary field and an incident one. Therefore, the resultant light of interference has wavelength dependence in intensity attenuation and the phase shift that occurs during propagation through an absorptive medium.

In the latter case that concerns the reflection of light from the SPP-generating interface, a reflection dip occurs as a result of destructive interference between a reflected part of an incident light and a leakage radiation induced by SPP evanescent back-coupling across the metal film thickness into a high refractive index layer (herein N-BK7 glass prism) [4,5]. This SPP radiation damping governed by the metal film thickness in turn modifies the real and the imaginary parts of the SPP propagation vector, which are all wavelength-dependent. When the leakage radiation with its wave vector also resembling that of SPP in its wavelength dependence interferes with a reflected part of an incident light, the dephasing degree between the two depends on a wavelength. In other words, the wavelength dependence of the complex SPP propagation vector determines the phase of the leakage radiation field. This eventually produces the wavelength dependence of both intensity and phase of the resultant light of interference, a feature similar to the case of an absorptive medium (the former case), as shown in Figure 4, the where spectral dependence of two phases are similar.

Figure 4 and Figure 5 also show that the mismatch between the two phases slightly grows as the wavelength goes far away from the resonance. This can arise from the fact that Equation (12) uses the wavelength-dependent OD(λ) over the limited wavelength range where the estimated $\epsilon_{r}^{\mathrm{Au}}\left(\lambda \right)$ is used for calculation. Table 1 shows a limited range used for OD calculation and the further limited range around the resonance where the two phases approximately coincide for an incident angle of 43.85°.

## 4. Conclusions

The dispersion of a phase shift of ATR light from a metal–dielectric interface is theoretically investigated using the KK relation that has its origin in physical causality. The wavelength dependence of optical reflectance from an SPP-generating interface at plasmonic resonance is assumed to correspond to the wavelength dependence of the absorption of light propagating through a virtual absorptive medium. Optical phase shifts calculated using the KK relation assuming a virtual absorptive medium and those calculated for reflected light from the SPP-generating interface using a field transfer matrix approach, both show good agreement with the wavelength dependence at around resonance.

It can thus be interpreted that physical causality dictates the wavelength-dependent phase of the plasmon-induced leakage radiation field (into the high index medium, i.e., a prism), which interferes with a reflected part of an incident light. It results in phase shift dispersion of the net reflected light (interfered sum of reflected part of an incident light and a leakage radiation) from the SPP interface. This is similar to the wavelength dependence of a phase shift of light through an absorptive medium where physical causality leads to the wavelength dependence of a phase of the secondary field induced by the medium polarization which interferes with a polarization-stimulating incident field of light.

Future work may include use of the KK relation to understand the properties of reflected light in an ATR scheme where another resonant system, such as a quantum dot, interacts with surface plasmon.

## Figures and Tables

**Figure 1 nanomaterials-11-03063-f001:**
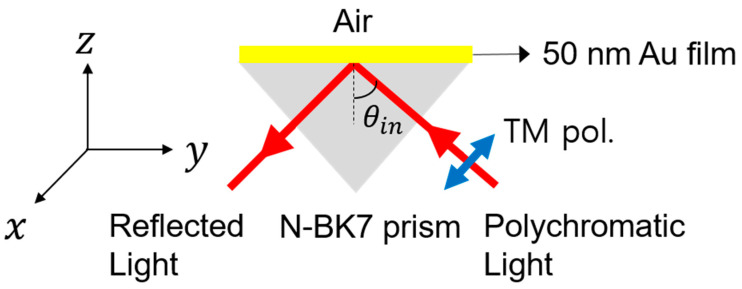
Schematic for ATR in a Kretschmann–Raether configuration to excite SPP at the Au film–air interface using TM polarization of light. Given an incident angle of light ($\theta_{in}$), reflectance reduces at a narrow spectral band as a result of SPP generation (wavelength interrogation method).

**Figure 2 nanomaterials-11-03063-f002:**
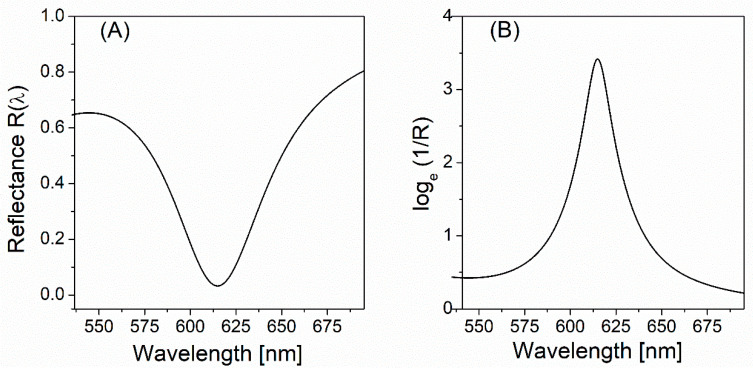
(**A**) The reflectance $R\left(\lambda \right)\equiv {\left|r\left(\lambda \right)\right|}^{2}$ calculated by a field transfer matrix-based formula (Equation (1)), assuming the Kretschmann–Raether configuration where SPP is excited at the interface between the 50 nm-thick Au film and air. The incident angle ($\theta_{in}$) is 43.85°. The reflection dip occurs at a wavelength of about 615 nm. (**B**) The natural logarithm of 1/R(λ) where R(λ) is given by (**A**).

**Figure 3 nanomaterials-11-03063-f003:**
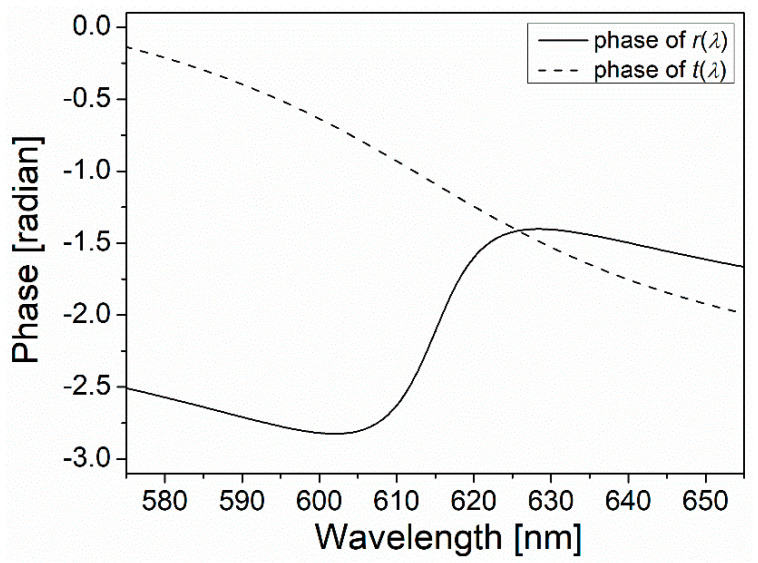
The wavelength dependence of the phase shift of the reflected and transmitted electric fields calculated using a field transfer matrix formula Equations (1) and (2), respectively.

**Figure 4 nanomaterials-11-03063-f004:**
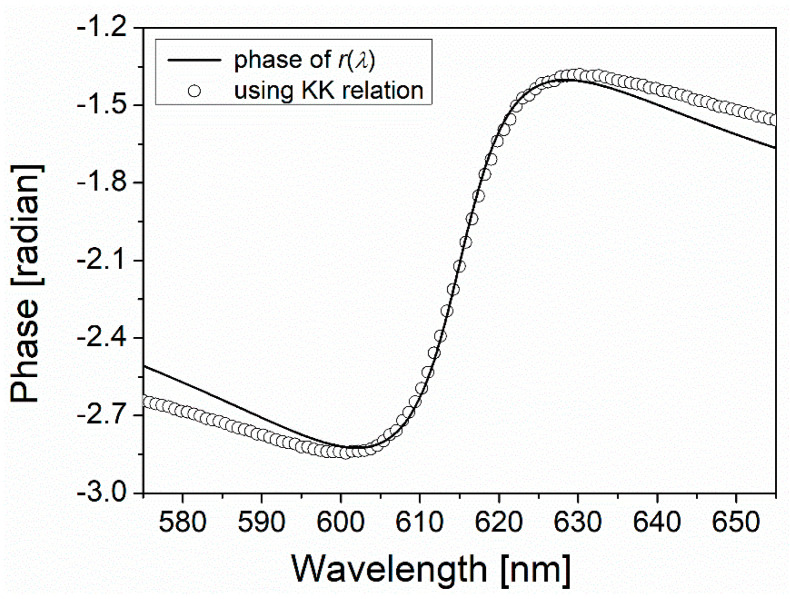
The wavelength-dependent phase calculated using the KK relation in comparison to the phase of r(λ) calculated from a field transfer matrix formula (Equation (1)). The incident angle ($\theta_{in}$) is 43.85°.

**Figure 5 nanomaterials-11-03063-f005:**
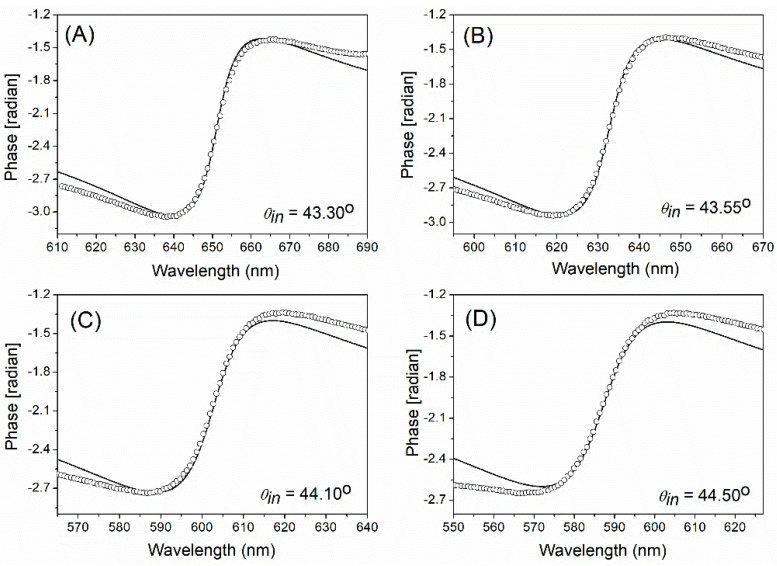
At other incident angles ($\theta_{in}$) of light, the phase calculated using the KK relation (empty circles) in comparison to the phase directly obtained from r(λ) calculated from a field transfer matrix formula Equation (1) (solid curves). The phase offsets used are $\phi_{0}=-2.28,-2.20,-2.06,-2.00$ radians for (**A**–**D**), respectively.

**Table 1 nanomaterials-11-03063-t001:** Wavelength ranges used for OD calculation and for two phases match.

	Wavelength Range
OD range (Figure 2B)	547.8–676.0 nm (OD max. at 615 nm)
Central range of matching in Figure 4	~600–~627 nm

## Data Availability

The data presented in this study are available on request from the corresponding author. The data are not publicly available due to privacy.

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
