# Peer review of "Kramers–Kronig Relation for Attenuated Total Reflection from a Metal–Dielectric Interface Where Surface Plasmon Polaritons Are Excited"

_nanomaterials, 2021, doi:10.3390/nano11113063_

Round 1

Reviewer 1 Report

Applicability of Kramers-Kronig relation to light reflected from the metal-dielectric interface that can excite surface plasmon polaritons (SPP) is theoretically investigated. The plasmon-induced attenuation of reflected light can be taken as resonant absorption of light through a virtual absorptive medium. Optical phase shift of light reflected from the SPP-generating interface is calculated using the KK-relation that uses spectral dependence of attenuation of light reflected from the interface at around the plasmonic resonance. This work is interesting and can be accepted after solving the following concerns.

  1. What’s the limit and assumption of using formula 11? Is it universal or only suitable in SPP structure?
  2. What’s the OD(λ) spectrum range used in the calculation of figure3? The table which describes the quantity relationship of center accuracy range and OD(λ) spectrum range could be given, to support the description in lines 199-202.
  3. Can the phase shift of transmitted light in the SPP structure be calculated and compared?
  4. The verificationwas bare on a 43.85-degree incident, please choose several different incident angles to verify the accuracy of the Kramers-Kronig relation phase shift calculation.
  5. Why italic type was used in lines 58-59 and 163-156? It seems not necessary.
  6. Some references that involve the dispersion of plasmon may help enrich the introduction, e.g., Chemical Society Reviews, 2019, 48(8): 2458-2494, Opto-Electron Adv 4, 210031 (2021), iscience, 2019, 21: 145-156.

Author Response

Replying Letter

Manuscript ID: nanomaterials-1451339

Title: Application of Kramers-Kronig Relation to Light Reflected from a Metal-dielectric Interface where Surface Plasmon Polaritons are Excited

Authors: Heongkyu Ju *

Thanks for your valuable comments and advice for the manuscript submitted to the journal “Nanomaterials”. Revisions have been made to address the points of concern raised by reviewers as underlined in the revised manuscript. Corresponding answers/responses to each question and comment have been provided as underlined as given below.

Reviewer 1

Applicability of Kramers-Kronig relation to light reflected from the metal-dielectric interface that can excite surface plasmon polaritons (SPP) is theoretically investigated. The plasmon-induced attenuation of reflected light can be taken as resonant absorption of light through a virtual absorptive medium. Optical phase shift of light reflected from the SPP-generating interface is calculated using the KK-relation that uses spectral dependence of attenuation of light reflected from the interface at around the plasmonic resonance. This work is interesting and can be accepted after solving the following concerns.

Thanks for your valuable comments and advice and please see the answers as below.

  1. What’s the limit and assumption of using formula 11? Is it universal or only suitable in SPP structure?

Answer: This equation (eq. (12) in the revised manuscript) can be derived from the equation (10) simply by multiplying it with a propagation distance through a medium and a wave number in vacuum. Equation (10), another form of the Kramers-Kronig relation under resonant absorption, relates the absorption coefficient (alpha) of a medium with the real part of the refractive index as also presented in references [J. Am. Chem. Soc. Vol. 128, pp. 10909; J. Am. Chem. Soc. 2007 Vol. 129, pp. 7647] (It does not suit only SPP structure). I have newly added a number of lines with the relevant references to explain how to derive this equation as seen in lines 154-158 of the revised manuscript.

  1. What’s the OD(λ) spectrum range used in the calculation of figure3? The table which describes the quantity relationship of center accuracy range and OD(λ) spectrum range could be given, to support the description in lines 199-202.

Answer: A table is added with relevant explanation to compare the OD wavelength range with the wavelength range of two phase matching around resonance as seen in lines 228-234 of the revised manuscript.

  1. Can the phase shift of transmitted light in the SPP structure be calculated and compared?

Answer: One more figure (Figure 3) has been added with a number of sentences to show the phase shift of the transmission coefficient to represent the relative phase shift of the transmitted field with respect to that of an incident field as seen in lines of 119-126 of the revised manuscript. In this figure, this phase is also compared to the reflection coefficient phase.

  1. The verification was bare on a 43.85-degree incident, please choose several different incident angles to verify the accuracy of the Kramers-Kronig relation phase shift calculation.

Answer: for other incidence angles, KK relation based phase shift also shows good agreement with that calculated using transfer matrix formula. Figures 5(A)-(D) with accompanying sentences have been added for four more number of different incident angles to verify the KK relation based calculation as seen in lines of 178-187 of the revised manuscript

.

  1. Why italic type was used in lines 58-59 and 163-156? It seems not necessary.

-

Answer: Italic types were made back to normal ones as seen in lines of 61-63, 129, 190-191 of the revised manuscript.

  1. Some references that involve the dispersion of plasmon may help enrich the introduction, e.g., Chemical Society Reviews, 2019, 48(8): 2458-2494, Opto-Electron Adv 4, 210031 (2021), iscience, 2019, 21: 145-156.

Answer: Some of the references recommended have been newly added [15-16] with relevant lines of sentences in the introduction section of the revised manuscript (see lines 56-58).

Reviewer 2

In this work, "Application of Kramers-Kronig relation to light reflected from a metal-dielectric interface where surface plasmon polaritons are excited", the authors theoretically investigated the applicability of KK relation to light reflected from the metal-dielectric interface that can excite SPP. Based on the obtained results, the authors claimed that the KK-relation-calculated phase shifts shows good agreement with that directly obtained from the reflection coefficient calculated by transfer matrix formula at around the resonance. Overall, this manuscript has a strong potential for a second review after applying the issues and addressing the shortcomings listed below:

Thanks for your valuable comments. Please see the answers as below.

1-The authors should polish/revise some grammatical mistakes and typos along the manuscript. I invite the authors to read their manuscript carefully and make the required changes where necessary.

Answer: I have revised the manuscript grammatically and made changes for better clarity. (See the underlined words/phrases and sentences).

2-If possible, try to revise the title of the manuscript.

Answer: I have changed to “Kramers-Kronig Relation for Attenuated Total Reflection from a Metal-dielectric Interface where Surface Plasmon Polaritons are Excited”

3-In the Introduction section, while discussing the recent developments in the field of surface plasmon resonances, the following works should be considered and cited to give a more general view to the possible readers of the work: [(i) Monolithic metal dimer-on-film structure: new plasmonic properties introduced by the underlying metal, Nano Letters 20, 2087-2093 (2020); (ii) The observation of high-order charge-current configurations in plasmonic meta-atoms: A numerical approach, Photonics 6, 43 (2019)].

Answer: I have added one of the recommended references (the first one) which is more relevant to the context of the introduction section. (see reference [17])

4-Corresponding references for Eq. 5 & 6 should be provided for the possible readers of the submitted manuscript.

Answer: I have added the references [1-3] for the two equations for the KK relation.

5-Any plan for the experimental verification of the extracted results? Please explain.

Answer: The current manuscript proposed another way of interpreting the phase shift and wavelength dependent intensity of reflected light in a ATR (attenuated total reflection) scheme, both of which have already been experimentally and theoretically estimated. Instead, I have added a number of lines for future works which may be interesting as related to the current manuscript work.

[Future work may include use of the KK relation to understand properties of reflected light in an ATR scheme where another resonant system such as a quantum dot interacts with surface plasmon.]

Reviewer 2 Report

In this work, "Application of Kramers-Kronig relation to light reflected from a metal-dielectric interface where surface plasmon polaritons are excited", the authors theoretically investigated the applicability of KK relation to light reflected from the metal-dielectric interface that can excite SPP. Based on the obtained results, the authors claimed that the KK-relation-calculated phase shifts shows good agreement with that directly obtained from the reflection coefficient calculated by transfer matrix formula at around the resonance. Overall, this manuscript has a strong potential for a second review after applying the issues and addressing the shortcomings listed below:

1-The authors should polish/revise some grammatical mistakes and typos along the manuscript. I invite the authors to read their manuscript carefully and make the required changes where necessary.

2-If possible, try to revise the title of the manuscript.

3-In the Introduction section, while discussing the recent developments in the field of surface plasmon resonances, the following works should be considered and cited to give a more general view to the possible readers of the work: [(i) Monolithic metal dimer-on-film structure: new plasmonic properties introduced by the underlying metal, Nano Letters 20, 2087-2093 (2020); (ii) The observation of high-order charge-current configurations in plasmonic meta-atoms: A numerical approach, Photonics 6, 43 (2019)].

4-Corresponding references for Eq. 5 & 6 should be provided for the possible readers of the submitted manuscript.

5-Any plan for the experimental verification of the extracted results? Please explain.

Author Response

(The authors gave the same response as above.)

Round 2

Reviewer 2 Report

In its current form, the revised manuscript is suitable for publication.